# Predictors of Discordance in the Assessment of Skeletal Muscle Mass between Computed Tomography and Bioimpedance Analysis

**DOI:** 10.3390/jcm8030322

**Published:** 2019-03-07

**Authors:** Min Ho Jo, Tae Seop Lim, Mi Young Jeon, Hye Won Lee, Beom Kyung Kim, Jun Yong Park, Do Young Kim, Sang Hoon Ahn, Kwang-Hyub Han, Seung Up Kim

**Affiliations:** 1Department of Internal Medicine, Institute of Gastroenterology, Yonsei University College of Medicine, Seoul 03722, Korea; zns-1@hanmail.net (M.H.J.); TSLIM21@yuhs.ac (T.S.L.); HYUK4385@yuhs.ac (M.Y.J.); LORRY-LEE@yuhs.ac (H.W.L.); BEOMKKIM@yuhs.ac (B.K.K.); DRPJY@yuhs.ac (J.Y.P.); DYK1025@yuhs.ac (D.Y.K.); AHNSH@yuhs.ac (S.H.A.); GIHANKHYS@yuhs.ac (K.-H.H.); 2Yonsei Liver Center, Severance Hospital, Seoul 03722, Korea; 3Institute of Gastroenterology, Severance Hospital Yonsei University College of Medicine, Seoul 03722, Korea

**Keywords:** sarcopenia, bioimpedance analysis, computed tomography, discordance

## Abstract

Computed tomography (CT) and bioimpedance analysis (BIA) can assess skeletal muscle mass (SMM). Our objective was to identify the predictors of discordance between CT and BIA in assessing SMM. Participants who received a comprehensive medical health check-up between 2010 and 2018 were recruited. The CT and BIA-based diagnostic criteria for low SMM are as follows: Defined CT cutoff values (lumbar skeletal muscle index (LSMI) <1 standard deviation (SD) and means of 46.12 cm^2^/m^2^ for men and 34.18 cm^2^/m^2^ for women) and defined BIA cutoff values (appendicular skeletal muscle/height^2^ <7.0 kg/m^2^ for men and <5.7 kg/m^2^ for women). A total of 1163 subjects were selected. The crude and body mass index (BMI)-adjusted SMM assessed by CT were significantly associated with those assessed by BIA (correlation coefficient = 0.78 and 0.68, respectively; *p* < 0.001). The prevalence of low SMM was 15.1% by CT and 16.4% by BIA. Low SMM diagnosed by CT was significantly associated with advanced age, female gender, and lower serum albumin level, whereas low SMM diagnosed by BIA was significantly associated with advanced age, female gender, and lower BMI (all *p* < 0.05). Upon multivariate analysis, age >65 years, female and BMI <25 kg/m^2^ had significantly higher risks of discordance than their counterparts (all *p* < 0.05). We found a significant association between SMM assessed by CT and BIA. SMM assessment using CT and BIA should be interpreted cautiously in older adults (>65 years of age), female and BMI <25 kg/m^2^.

## 1. Introduction

Sarcopenia is a syndrome characterized by the loss of skeletal muscle mass, strength, and performance [1,2,3] that results in an increased risk of fracture, dysfunction, reduced quality of life, and increased mortality [4,5]. Due to the varying diagnostic cutoff values for muscle mass and the varying diagnostic tools used in previous studies, the reported prevalence of sarcopenia has been inconsistent [6,7]. Several studies of sarcopenia have been performed, and multiple guidelines have been proposed; these have enhanced our knowledge of the condition. Sarcopenia is now officially recognized as a disorder in some countries, with an ICD-10-MC diagnostic code [8].

The measurement of skeletal muscle mass (SMM) is of paramount importance to diagnose sarcopenia. Several imaging techniques that can assess SMM are currently available: Dual-energy X-ray absorptiometry (DXA), computed tomography (CT), magnetic resonance imaging (MRI), and bioimpedance analysis (BIA) [2]. DXA has several advantages over the other methods, such as safety, accuracy, and non-invasiveness, but it can overestimate muscle mass in cases of muscle edema or intramuscular fat deposition [9]. CT can accurately measure the quantity and quality of SMM, but it is costly and exposes the patient to radiation [2,10]. MRI has no radiation exposure for the patient and accurately measures SMM, but its clinical application is significantly limited due to its high cost [11]. BIA has been recognized as a rapid, inexpensive, portable, and safe methodology but, because BIA measures the resistance to a current that is applied through a body of water, the assessment of SMM may be inaccurate if the patients are dehydrated, overhydrated fluid status or obese [12]. BIA tends to overestimate SMM because it cannot discriminate among appendicular, non-appendicular fat, and non-fat mass [13].

To date, measurement of SMM by BIA has typically been performed using the Kyle, Jassen, Ergi, and Scafoglieri prediction models [14,15,16,17]. SMM can now be assessed directly using vertical, eight-point analyzers. Several studies have reported the accuracy and reproducibility of direct segmental multi-frequency BIA and the strong correlation between its results and SMM measured by DXA [18,19,20,21,22,23]. CT provides an accurate measurement of SMM, with a significant correlation to whole-body muscle mass [24,25]. Accordingly, CT has been considered to be the gold standard for measuring SMM [2,10]. Despite its drawbacks, BIA is more easily applied in clinical practice. Thus, investigating the prevalence of discordance in the assessment of SMM between CT and BIA and identifying predictors of this discordance is valuable. This investigation can ultimately help physicians select the optimal candidates for each modality to diagnose low SMM and interpret the results appropriately.

The primary aim of this study was to identify predictors of discordance between SMM measured by BIA and by CT. The secondary aims were to investigate the prevalence of low SMM by CT and BIA and the correlation between SMM measured by CT and BIA in apparently healthy subjects undergoing comprehensive medical health check-ups.

## 2. Methods

### 2.1. Study Subjects

A total of 1191 subjects who visited the health promotion center in Severance Hospital, a university-affiliated tertiary care hospital, for a comprehensive medical health check-up from June 2010 to April 2018 were included, see Figure 1. Severance Hospital is a 2000-bed academic referral hospital in Northwestern Seoul, Republic of Korea. Severance Hospital is supported by Yonsei University College of Medicine. Exclusion criteria were as follows: (1) no BIA data, (2) limited access to BIA data due to personal privacy, (3) poor CT quality, and (4) major operation in the lumbar area. 

The study’s protocol adhered to the tenets of the Declaration of Helsinki and was approved by the Institutional Review Board of Severance Hospital. Informed consents were waived due to the retrospective nature of the study.

### 2.2. Data Collection

A medical health check-up was performed, and collected data included age, gender, height, body weight, body mass index (BMI), and laboratory test results. Histories of hypertension, diabetes, and viral hepatitis were collected from the medical record and individual questionnaires.

### 2.3. Fibrosis-4 Index Calculation

Recent studies have shown that fibrotic burden in the liver is independently associated with sarcopenia. Therefore, the fibrosis-4 index (FIB-4) was calculated using the following formula: Age (years) × aspartate aminotransferase (AST) (U/L)/(platelets (10^9^/L) × alanine aminotransferase (ALT) (U/L])^1/2^ [26].

### 2.4. Measurements of Skeletal Muscle Area

Skeletal muscle area was measured at the mid-body level of the L3 vertebra in a supine position by a dual-source 128-slice CT scanner (Somatom Definition Flash, Siemens Healthcare, Forchheim, Germany), a 64-slice CT scanner (Somatom Sensation 64, Siemens Healthcare), a Discovery 710 PET-CT 128-slice scanner (General Electric Medical Systems, Milwaukee, WI, USA), a Biograph TruePoint 40 PET-CT 40-slice scanner (Siemens Medical Solutions, Hoffman Estates, IL, USA), or a Discovery 600 PET-CT 16-slice scanner (General Electric Medical Systems, Milwaukee, WI, USA). The muscle area was identified using attenuation values between −29 to −150 Hounsfield units. Total lumbar skeletal muscle area (psoas, erector spinae, quadratus lumborum, transversus abdominus, external and internal obliques, and rectus abdominus) (cm^2^) was defined as a region with density ranging from −29 to −150 Hounsfield units using Aquarius Intuition Viewer software, version 4.4.12 (Terarecon, San Mateo, CA, USA). Boundaries were corrected manually, as necessary. To minimize measurement error, the CT instruments are periodically tested and calibrated for spatial resolution, length measurement, alignment, and linearity of attenuation (CT number) using a standard phantom (AAPM CT Performance Phantom, 76-410). All tests are performed in compliance with the regulations of the Korean Institute for Accreditation of Medical Imaging. The lumbar skeletal muscle index (LSMI) was defined as 10,000 × lumbar skeletal muscle area (LSMA, cm^2^)/height^2^ (m^2^). Based on previous studies [2,10], we assumed that measurement of SMM by CT is more accurate.

The InBody 770 (Biospace Co., Seoul, Korea) measured body composition. Participants fasted for 12 h prior to testing. Participants wore a t-shirt and short pants on the day of testing, and provided their age, gender, and height at the time of measurement. Testing was conducted according to the manufacturer’s instructions. Data were uploaded to the electronic medical record. The measurement was comprised of two combinations: z-axis at frequencies of 1, 5, 50, 250, and 500 kHz for impedance, and x-axis at frequencies of 5, 50, and 250 kHz for reactance. Impedance was measured for five body segments: Trunk, right and left arms, and right and left legs. We reviewed the medical records and measured appendicular skeletal muscle (ASM) (kg) through direct segmental multi-frequency BIA [27].

Muscle mass was determined by measuring electrical resistance [28] using four surface tactile electrodes placed on the dorsal surface of the hand and foot. Whole-body resistance (R_sumx_) was calculated by summing the segmental resistances at frequency x, according to the following equation:R_sumx_ = R_RA_ + R_LA_ + R_T_ + R_RL_ + R_LL_(1)

The index of R_sumx_ (RI_sumx_) is calculated by using the following equation:RI_sumx_ Height (cm)^2^/R_sumx_ (Ω)(2)
Appendicular muscle mass = 0.236 × Height^2^/R_RA_ + 0.0109 × Hright^2/^R_T_ + 0.121 × Hright^2^/R_RL_ +1.554(3)

Using the formula above, the muscle mass is automatically calculated in InBody.

### 2.5. Definition of Low SMM

The CT diagnostic criterion for low SMM was a lumbar skeletal mass index (LSMI) <1 standard deviation (SD) below the sex-specific mean of the study group. The BIA diagnostic criterion for low SMM was adopted from the Asian Working Group of Sarcopenia [6]: ASM/height^2^ <7.0 kg/m^2^ for men and <5.7 kg/m^2^ for women [29].

We also used additional CT and BIA diagnostic criteria for low SMM. The additional criterion for CT was an LSMI ≤52.4 cm^2^/m^2^ for men and ≤38.5 cm^2^/m^2^ for women [30]. The additional criterion for BIA was adopted from The Foundation for the National Institutes of Health: ASM/BMI <0.79 for men and <0.51 for women [7]. We attached the relevant analysis using Appendix A.

### 2.6. Statistical Analysis

Statistical analyses were performed using Statistical Package for the Social Science (SPSS) version 23.0 for Windows (IBM Corp., Armonk, NY, USA). Continuous and categorical variables were expressed as mean ± standard deviation and *n* (%), respectively. *p*-Value < 0.05 was considered statistically significant. Simple and partial correlation analyses were used to analyze the relationship between CT and BIA muscle mass. The distribution between muscle mass by BIA and quartile stratification of muscle mass by CT was evaluated using the Mann-Whitney U test. The comparison between subjects with and without low SMM was performed using the chi-square test for categorical variables and Student’s t-test for continuous variables. Multivariate analysis using binary logistic regression analysis was performed on variables that showed a *p*-value <0.05 and was used to determine the predictors of discordance in defining low SMM between CT and BIA.

## 3. Results

### 3.1. Patients

A total of 1191 subjects who underwent a comprehensive medical health check-up were considered eligible. However, 19 subjects were excluded due to a lack of BIA data, and an additional nine subjects were excluded due to poor-quality CT scans and a history of a major operations around the lumbar or appendicular skeletal muscle area. As a result, 1163 subjects were included in the statistical analysis, see Appendix A.

Baseline characteristics of the study population (641 men and 521 women) are summarized in Table 1. The mean age of the patients was 57 years; 41.0% were over 60 years of age. The mean BMI of the patients was 24.0 kg/m^2^. Of the study population, 41.0% of subjects (*n* = 488) had hypertension, 29.4% (*n* = 314) had diabetes, and 4.9% (*n* = 57) had viral hepatitis. Using CT scans, the mean whole-body fat-free mass and LSMI were 45.3 kg and 46.9 cm^2^/m^2^, respectively. Using BIA, the mean ASM, ASM index, and ASM/BMI ratio were 20.1 kg, 7.1 kg/m^2^, and 0.82, respectively. The mean FIB-4 was 1.17.

### 3.2. Association between SMM Assessed Using CT and BIA

The crude and BMI-adjusted SMM assessed by CT were significantly associated with those assessed by BIA (*p* < 0.001, correlation coefficient = 0.898 for crude SMM; *p* < 0.001, correlation coefficient = 0.858 for BMI-adjusted SMM), see Figure 1A. The association between crude SMM assessed by CT and BIA was statistically significant, regardless of gender (*p* < 0.001, correlation coefficient = 0.724 in men; *p* < 0.001, correlation coefficient = 0.645 in women), as shown in Figure 1. Linear regression results comparing CT and BIA assessed SMM were added to the Appendix A.

We divided the patients into four groups according to quartiles of SMM assessed by CT and BIA. SMM as assessed by BIA significantly increased according to the CT-assessed SMM quartile (*p* < 0.001), see Appendix A.

### 3.3. Comparison between Subjects with and without Low SMM Assessed by CT

The baseline characteristics of subjects with and without CT-defined low SMM in Table 2. The cutoff value of low SMM was defined as less than one standard deviations sex-specific mean value of the participants. The sex-specific cut-off values of LSMI were 46.12 cm^2^/m^2^ in men and 34.18 cm^2^/m^2^ in women.

When CT-defined cutoff values were used, subjects with low SMM were significantly older (median 63 vs. 57 years) and female gender (48.8% vs. 44.1%). Subjects with low SMM had significantly lower serum albumin levels (median 4.2 vs. 4.3 mg/dL), lower total cholesterol (median 177 vs. 188 mg/dL), higher high-density lipoprotein (HDL) cholesterol (median 50 vs. 48 mg/dL) and lower low-density lipoprotein (LDL) cholesterol (median 100 vs. 111 mg/dL) than those of subjects without low SMM (all *p* < 0.05). In addition, various muscle indexes were unfavorable in subjects with CT-defined low SMM.

We also analyzed additional diagnostic criteria for low SMM defined by CT (≤52.4 cm^2^/m^2^ for men and ≤ 38.5 cm^2^/m^2^ for women), see Appendix A.

### 3.4. Comparison between Subjects with and without Low SMM Assessed by BIA

The baseline characteristics of subjects with and without BIA-defined low SMM are shown in Table 3. The cutoff value of low SMM was defined previous study [6]. The Asian Working Group of Sarcopenia defined cutoff values appendicular lean mass (ALM)/height^2^ of <7.0 kg/m^2^ in men and <5.7 kg/m^2^ in women.

When BIA-defined cutoff values were used, subjects with low SMM were significantly older (median 60 vs. 57 years) and had a higher proportion of female subjects (67.5% vs. 40.0%), lower BMI (median 21.8 vs. 24.2 kg/m^2^) (all *p* < 0.05). In addition, various muscle indices were unfavorable in subjects with BIA-defined low SMM.

We also analyzed additional diagnostic criteria for low SMM defined by BIA (ALM/BMI <0.79 for men and <0.51 for women), see Appendix A.

### 3.5. Prevalence and Predictors of Discordance in Defining Low SMM Assessed by CT and BIA

The proportion of non-discordant and discordant subjects, when different measuring methods were applied (CT vs. BIA), is described in Table 4. The proportion of subjects without low SMM by both CT and BIA was 72.3%, and that of subjects with low SMM ranged was 3.9%. The overall proportion of non-discordant subjects was 76.2%. The results of analysis using additional diagnostic criteria for low SMM are given in Appendix A.

To identify the predictors of discordant results by CT and BIA, univariate analysis was performed, see Table 5. Older age (HR = 1.05), female sex (HR = 1.48), lower BMI (HR = 0.73), lower serum albumin level (HR = 0.58), and higher GGT (HR = 1.01) were significantly predictive of discordance between CT- and BIA-defined low SMM (*p* < 0.05). The results of analyses using the additional diagnostic criteria for low SMM are listed in Appendix A. The results of basic demographic characteristics, specificity and sensitivity of the low SMM defined by the BIA compared to the low SMM defined by the CT as diagnostic standard criteria, added to Appendix A.

Among selected independent predictors of the presence of discordance, age, female gender, and BMI were selected for multivariate analysis. Thus, we stratified our study population into two groups according to these three independent variables to check the prevalence of discordance, see Figure 2. Older age (>65 years) (22.3% vs. 12.2%), female gender (20.9% vs. 9.8%), and lower BMI (<25 kg/m^2^) (20.1%% vs. 3.5%) had a significantly higher risk of discordance than the counterparts (all *p* < 0.001). The results of analysis using additional diagnostic criteria for low SMM are given in Appendix A. 

## 4. Discussion

The diagnostic criteria for sarcopenia have not yet been definitively established, even though it is one of the most important public health concerns [31]. Varying diagnostic criteria for sarcopenia based on several assessment modalities, which include CT and BIA, are available [2,3,10], and the criteria are different between Asian and Western countries [6,7,32]. Ethnicity is an important factor for the diagnosis of sarcopenia [33]. Several recent research groups have published diagnostic guidelines for sarcopenia, which have emphasized the importance of ethnicity [34,35]. The BIA and CT diagnostic criteria differ according to ethnicity [3,6,7,35]. According to our knowledge, no comparison of BIA and SMM measured by CT at the L3 level has been performed. Therefore, our findings will facilitate the establishment of diagnostic cutoff values for Asian patients.

Our data show a significant association in crude and BMI-adjusted SMM assessed by CT and BIA, although the area assessed was different for each method. Similar to previous studies [3,11,32,36,37,38,39], the proportion of subjects with low SMM in our study varied from 15.1% to 16.4% when CT or BIA was used to assess SMM, and the risk factors for discordant results between the methodologies were advanced age, female gender, and lower BMI.

We believe the identified risk factors for discordant results can be explained in several ways: Total fat mass tends to be higher in older adults [40], and BIA can overestimate SMM when the subject has a high fat mass [41]; assessment of SMM using BIA can be overestimated in female subjects who have a higher probability of increased body fat [41]; and there is a weaker correlation between SMM in the limb and L3 area among subjects with a lower BMI [42,43]. All of these factors suggest that CT may be required for a more accurate assessment of SMM in subjects with advanced age, female gender, and low BMI.

Our study has several strengths. First, the overall sample size was over 1100, which ensures the statistical power and precision of our results. We adopted several cutoff values for CT and BIA when defining low SMM, and we found the three factors of age, gender, and BMI to be associated with discordance between CT- and BIA-based SMM assessments. Second, we focused on the general population instead of medically vulnerable subjects, such as only older adults, or those with liver cirrhosis or cancers for whom sarcopenia already showed clinical implications. Similar to our study, several recent studies proved the clinical significance of assessing sarcopenia in the general population and non-alcoholic fatty liver disease (NAFLD) subjects [44,45]. Thus, our study provides information that helps to identify optimal subjects for CT-based assessment of sarcopenia. Third, in contrast to most previous studies [7,11,27,46], we adopted several cutoff values for SMM assessed by CT and BIA. Although the predictors of discordance were not exactly the same, we obtained relatively consistent results regardless of the cutoff value used. Fourth, several studies have compared DXA and BIA, but few have directly compared CT and BIA to assess SMM [46,47]. In our study, SMM using CT and BIA was measured on the same day, in contrast to most previous studies [9,25,46]. As a result, any bias caused by different time points of SMM assessments may have been prevented. Lastly, because there are significant differences in SMM between Western and Asian populations, focusing on a single ethnicity may be important. Thus, the results of our study could be optimized for an Asian population.

Several issues remain unresolved in our study. First, although we adopted several known cutoff values for CT and BIA, the results of our study should be further validated based on existing diagnostic criteria for sarcopenia. Second, recent studies have insisted that other factors, such as muscle strength and walking speed, should be considered when diagnosing sarcopenia. However, our study was retrospectively performed based on the clinical information of the subjects who underwent a comprehensive medical health check-up, and we only used SMM to define sarcopenia. Further studies with additional markers of sarcopenia should validate our results. Third, our study only included subjects who were willing to receive and could afford a medical health check-up. The prevalence of hypertension (29.1%) and diabetes (11.3%) in the general Korean population in 2016 (Korean Center for Disease Control and Prevention; Ministry of Health and Welfare) [48], were lower than those in this study (41.0% and 26.4%, respectively). In Korea, individuals >40 years of age are eligible for basic health check-ups; those with chronic diseases such as hypertension and diabetes receive health checkups more frequently. The mean age of our patients was 57 years, and 40.9% were >60 years of age. Because of this potential selection bias, our results may not be fully applicable to the general population, but this can be resolved in future studies. Fourth, SMM measured by BIA is affected by the hydration status [12]. The patients were admitted to the health check-up unit and fasted overnight. Thus, the hydration status of all of the patients should have been similar. Lastly, when discordant results between CT- and BIA-based SMM assessments were obtained, we did not know which diagnostic modality to accept. For patients with discordant CT and BIA results, it is important to decide which results should be used. However, a definitive diagnostic method for sarcopenia has not been established. This issue should be explored in future longitudinal follow-up studies that use solid end-points, such as mortality. This issue should be explored in future longitudinal follow-up studies that use solid end-points such as mortality, which might propose the right direction toward CT or BIA.

In conclusion, the significant association between CT and BIA for SMM assessment suggests that BIA could be used to assess sarcopenia in clinical practice. However, because advanced age, female gender, and low BMI were risk factors for discordant results between CT and BIA, BIA assessment should be interpreted cautiously in subjects with these risk factors and, if possible, CT or other modalities should be considered as an alternative diagnostic tool to assess SMM to define sarcopenia.

## Figures and Tables

**Figure 1 jcm-08-00322-f001:**
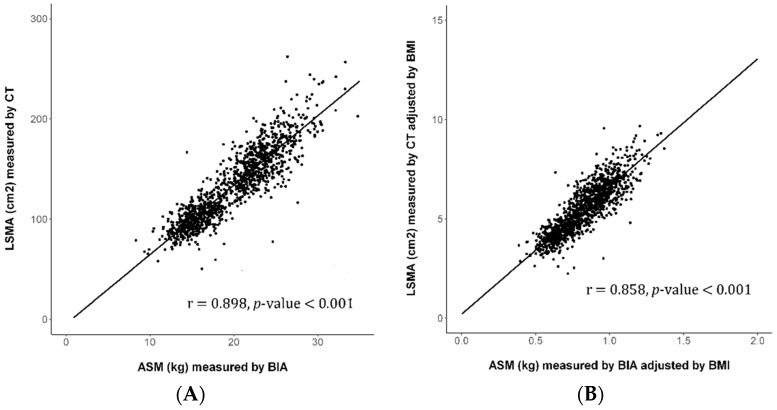
Correlation between SMM assessed by CT and BIA. Crude (**A**) and BMI-adjusted SMM (**B**) assessed by CT were significantly correlated with those by BIA (all *p* < 0.001, correlation coefficient = 0.898 and 0.858, respectively). The correlation between crude SMM assessed by CT and BIA was significant in men and women (**C**) (all *p* < 0.001, correlation coefficient = 0.724 in men and 0.645 in women, respectively). LSMA (cm^2^) = −4.366 + 6.920 * ASM (kg), Standard error = 0.099 LSMI adjusted by BMI = 0.212 + 6.424 * (ASM adjusted by BMI), Standard error = 0.113. SMM, skeletal muscle mass; LSMA, lumbar skeletal mass area; AMS, appendicular skeletal mass; CT, computed tomography; BIA, bioimpedance analysis; BMI, body mass index; LMSI, lumbar skeletal muscle index. Regression equations and standard error are as follows.

**Figure 2 jcm-08-00322-f002:**
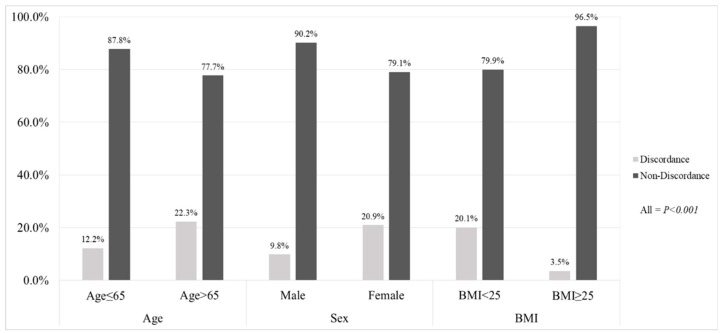
Percentage of subjects with non-discordance and those with discordance in diagnosing low SMM using CT and BIA according to identified independent predictors. Participants with age > 65 years, female gender and BMI < 25 kg/m^2^ had a significantly higher proportion of discordance than the counterparts (all *p* < 0.001). CT cutoff indicates < 1 SD. BIA cutoff indicates AWGS index (ASMI, ALM/height^2^) of <7.0 kg/m^2^ in men and <5.7 kg/m^2^. BMI, body mass index; ALM, appendicular lean mass; ASMI, appendicular skeletal mass index; AWGS, Asian Working Group of Sarcopenia.

**Table 1 jcm-08-00322-t001:** Baseline characteristics (*n* = 1163).

Variables	All
Demographic parameters	
Age, years	57 (18–92)
<40	59 (5.0)
40–49	182 (15.6)
50–59	445 (38.2)
60–69	296 (25.4)
>70	181 (15.5)
Female gender	521 (43.7)
Body mass index, kg/m^2^	24.0 (15.4–43.9)
Hypertension	488 (41.0)
Diabetes mellitus	314 (26.4)
Viral hepatitis	57 (4.9)
Laboratory parameters	
Fasting glucose, mg/dL	96 (58–340)
Aspartate aminotransferase, IU/L	21 (8–140)
Alanine aminotransferase, IU/L	19 (3–196)
Serum albumin, mg/dL	4.3 (3.4–5.3)
Total bilirubin, mg/dL	0.7 (0.2–4.0)
Gamma glutamyl-transpeptidase, IU/L	23 (6–539)
Serum creatinine, mg/dL	0.8 (0.4–7.3)
Platelet count, 10^9^/L	231 (89–846)
Prothrombin time, INR	0.9 (0.7–2.3)
Total cholesterol, mg/dL	187 (83–392)
Triglycerides, mg/dL	103 (31–815)
High-density lipoprotein cholesterol, mg/dL	48 (23–115)
Low-density lipoprotein cholesterol, mg/dL	109 (27–299)
HbA1c, %	5.8 (4.4–13.4)
Fibrosis-4 index	1.17 (0.20–5.47)
Muscle mass parameters	
By computed tomography	
Whole body fat-free mass, kg	45.3 (21.2–84.7)
Lumbar skeletal muscle index, cm^2^/m^2^	46.9 (20.0–85.6)
By bioimpedance analysis	
ASM, kg	20.1 (8.3–34.7)
ASM index, kg/m^2^	7.1 (3.2–28.9)
ASM/body mass index	0.82 (0.43–1.23)

Variables are expressed as median (interquartile range) or *n* (%). INR, international normalized ratio; ASM, appendicular skeletal muscle mass; ASMI, appendicular skeletal mass index.

**Table 2 jcm-08-00322-t002:** Comparison between subjects with and without low SMM assessed by CT.

Variables	without Low SMM	with Low SMM	*p*-Value
	(*n* = 988, 84.9%)	(*n* = 176, 15.1%)	
Demographic parameters			
Age, years	57 (19–92)	63 (18–92)	0.001
Female gender	435 (44.1)	86 (48.8)	0.017
Body mass index, kg/m^2^	24.2 (16.5–43.8)	22.4 (15.4–28.9)	0.584
Hypertension	411 (41.5)	77 (43.7)	0.436
Diabetes mellitus	268 (27.1)	46 (26.1)	0.780
Viral hepatitis	45(4.5)	12 (6.8)	0.152
Laboratory parameters			
Fasting glucose, mg/dL	96 (58–340)	96 (65–325)	0.820
Aspartate aminotransferase, IU/L	21 (8–140)	20 (11–69)	0.964
Alanine aminotransferase, IU/L	19 (3–196)	18 (4–58)	0.756
Serum albumin, mg/dL	4.3 (3.4–5.3)	4.2 (3.5–4.8)	0.025
Total bilirubin, mg/dL	0.7 (0.2–4.0)	0.7 (0.2–2.8)	0.441
Gamma glutamyl-transpeptidase, IU/L	23 (7–398)	22 (6–539)	0.407
Serum creatinine, mg/dL	0.81 (0.38–7.3)	0.74 (0.41–2.74)	0.828
Platelet count, 10^9^/L	232 (89–846)	229 (122–438)	0.654
Prothrombin time, INR	0.93 (0.78–2.28)	0.94 (0.73–2.15)	0.574
Total cholesterol, mg/dL	188 (83–392)	177 (98–302)	0.007
Triglycerides, mg/dL	105 (31–684)	84.5 (43–815)	0.825
High-density lipoprotein cholesterol, mg/dL	48 (24–100)	50 (23–115)	0.027
Low-density lipoprotein cholesterol, mg/dL	111 (27–299)	100 (43–213)	0.038
HbA1c, %	5.8 (4.4–13.4)	5.8 (4.7–12.4)	0.436
Fibrosis-4 index	1.15 (0.20–5.47)	1.31 (0.37–3.39)	0.825
Muscle mass parameters			
By computed tomography			
Whole body fat-free mass, kg	46.4 (23.9–84.7)	40.8 (21.2–79.3)	0.530
Lumbar skeletal muscle index, cm^2^/m^2^	48.7 (34.2–85.6)	39.9 (20.0–46.0)	<0.001
By bioimpedance analysis			
ASM, kg	20.3 (10.3–34.7)	18.1 (8.3–27.6)	0.973
ASM index, kg/m^2^	7.30 (4.62–10.58)	6.39 (3.24–8.64)	<0.001
ASM/body mass index	0.81 (0.45–1.23)	0.82 (0.43–1.17)	0.044

Variables are expressed as median (interquartile range) or *n* (%). SMM, skeletal muscle mass; CT, computed tomography; INR, international normalized ratio; ASM, appendicular skeletal muscle mass; ASMI, appendicular skeletal mass index. * CT cutoff indicates <1 standard deviation (SD), sex–specific mean value of the participants.

**Table 3 jcm-08-00322-t003:** Comparison between subjects with and without low SMM assessed by BIA.

Variables	without Low SMM	with Low SMM	*p*-Value
	(*n* = 972, 83.6%)	(*n* = 191, 16.4%)	
Demographic parameters			
Age, years	57 (19–92)	60 (18–92)	<0.001
Female gender	392 (40.0)	129 (67.5)	<0.001
Body mass index, kg/m^2^	24.2 (17.1–43.8)	21.8 (15.4–27.8)	0.005
Hypertension	411 (42.2)	77 (40.3)	0.474
Diabetes mellitus	273 (28.0)	41 (21.4)	0.339
Viral hepatitis	48 (4.9)	9 (4.7)	0.757
Laboratory parameters			
Fasting glucose, mg/dL	97 (58–340)	94 (65–265)	0.701
Aspartate aminotransferase, IU/L	21 (8–140)	20 (11–69)	0.604
Alanine aminotransferase, IU/L	20 (3–196)	17 (5–66)	0.683
Serum albumin, mg/dL	4.3 (3.4–5.2)	4.2 (3.5–5.3)	0.135
Total bilirubin, mg/dL	0.7 (0.2–4.0)	0.7 (0.3–2.5)	0.740
Gamma glutamyl-transpeptidase, IU/L	23 (6–398)	19 (7–539)	<0.001
Serum creatinine, mg/dL	0.82 (0.38–7.01)	0.69 (0.39–7.3)	0.016
Platelet count, 10^9^/L	230 (89–846)	241 (122–458)	0.256
Prothrombin time, INR	0.93 (0.73–2.28)	0.94 (0.78–2.15)	0.027
Total cholesterol, mg/dL	185 (83–392)	194 (98–300)	0.918
Triglycerides, mg/dL	106 (31–815)	88 (36–435)	0.915
High-density lipoprotein cholesterol, mg/dL	47 (23–98)	53 (29–115)	0.082
Low-density lipoprotein cholesterol, mg/dL	108 (27–299)	112 (43–213)	0.968
HbA1c, %	5.8 (4.7–13.4)	5.8 (4.4–10.5)	0.386
Fibrosis-4 index	1.16 (0.32–5.47)	1.29 (0.20–4.82)	<0.001
Muscle mass parameters			
By computed tomography			
Whole body fat-free mass, kg	47.7 (21.2~84.7)	34.4 (23.5~54.2)	0.131
Lumbar skeletal muscle index, cm^2^/m^2^	48.5 (20.0–85.6)	38.0 (27.0–58.1)	<0.001
By bioimpedance analysis			
ASM, kg	21.3 (12.8–34.7)	14.0 (8.3–21.6)	<0.001
ASM index, kg/m^2^	7.45 (5.70–10.58)	5.56 (3.24–6.99)	<0.001
ASMI/body mass index	0.83 (0.45–1.23)	0.76 (0.43–1.08)	0.568

Variables are expressed as median (interquartile range) or *n* (%). BIA Cutoff indicates AWGS index. SMM, skeletal muscle mass; BIA, bioimpedance analysis; INR, international normalized ratio; FNIH, The Foundation for the National Institutes of Health; ALM, appendicular lean mass; ASMI, appendicular skeletal mass index; BMI, body mass index; AWGS, Asian Working Group of Sarcopenia.

**Table 4 jcm-08-00322-t004:** Distribution of subjects with and without low SMM assessed by CT and BIA.

Muscle Mass Assessed by CT	Muscle Mass Assessed by BIA	
	* BIA Cutoff	
	without Low SMM	with Low SMM
	(*n* = 972, 83.6%)	(*n* = 191, 16.4%)
** CT cutoff		
Without low SMM (*n* = 987, 84.9%)	841 (72.3)	146 (12.6)
With low SMM (*n* = 176, 15.1%)	131 (11.3)	45 (3.9)

Variables are expressed as *n* (%). * BIA cutoff indicates AWGS index (ASMI, ALM/height^2^) of <7.0 kg/m^2^ in men and <5.7 kg/m^2^. ** CT cutoff indicates <1 SD, sex-specific mean value of the participants. SMM, skeletal muscle mass; CT, computed tomography; BIA, bioimpedance analysis; FNIH, The Foundation for the National Institutes of Health; ALM, appendicular lean mass; ASMI, appendicular skeletal mass index; BMI, body mass index; AWGS, Asian Working Group of Sarcopenia.

**Table 5 jcm-08-00322-t005:** Predictors of discordance between CT and BIA-based low SMM.

Variables	Discordance between CT and BIA-Based Low SMM		
	Univariate	Multivariate	
	*p*-Value	*p*-Value	OR (95% CI)
Demographic parameters			
Age, years	<0.001	<0.001	1.050 (1.035–1.069)
Female gender	<0.001	0.044	1.480 (1.012–2.303)
Body mass index, kg/m^2^	<0.001	<0.001	0.725 (0.668–0.790)
Hypertension	0.420	-	-
Diabetes mellitus	0.219	-	-
Viral hepatitis	0.875	-	-
Laboratory parameters			
Fasting glucose, mg/dL	0.728	-	-
Aspartate aminotransferase, IU/L	0.734	-	-
Alanine aminotransferase, IU/L	0.092	0.021	0.977 (0.955–0.996)
Serum albumin, mg/dL	0.020	0.097	0.581 (0.264–1.117)
Total bilirubin, mg/dL	0.691	-	-
Gamma glutamyl-transpeptidase, IU/L	0.002	<0.001	1.009 (1.004–1.014)
Serum creatinine, mg/dL	0.803	-	-
Platelet count, 10^9^/L	0.759	-	-
Prothrombin time, INR	0.084	0.153	3.580 (0.629–19.312)
Total cholesterol, mg/dL	0.277	-	-
Triglycerides, mg/dL	0.096	0.128	1.003 (0.999–1.005)
High-density lipoprotein cholesterol, mg/dL	0.096	0.280	1.011 (0.993–1.023)
Low-density lipoprotein cholesterol, mg/dL	0.172	-	-
HbA1c, %	0.696	-	-
Fibrosis-4 index	0.615	-	-

CT cutoff indicates <1 SD, sex-specific mean value of the participants. BIA cutoff indicates AWGS index (ASMI, ALM/height^2^) of <7.0 kg/m^2^ in men and <5.7 kg/m^2^. SMM, skeletal muscle mass; CT, computed tomography; BIA, bioimpedance analysis; OR, odds ratio; CI, confidence interval; INR, international normalized ratio; FNIH, The Foundation for the National Institutes of Health; ALM, appendicular lean mass; ASMI, appendicular skeletal mass index; BMI, body mass index; AWGS, Asian Working Group of Sarcopenia.

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
