# Peer review of "Predictors of Discordance in the Assessment of Skeletal Muscle Mass between Computed Tomography and Bioimpedance Analysis"

_jcm, 2019, doi:10.3390/jcm8030322_

Round 1
Reviewer 1 Report
Thank you for addressing the points put forward in the previous review. There are a couple more edits I think need attention.
page 2 57-59 - you state that the new device does not need to use prediction equations, yet on page 3 line 131 you detail a prediction equation for the calculation of muscle mass.
page 2 line 64 to 68. I'm not sure how these results can help a physician decide which modality to use for a particular patient? Your conclusion states that this still needs to be resolved in further studies.
Author Response
Comment 1
page 2 57-59 - you state that the new device does not need to use prediction equations, yet on page 3 line 131 you detail a prediction equation for the calculation of muscle mass.
Response>
Thank you for your comments. We just meant that SMM assessment can be done automatically. To avoid misunderstanding, we modified Introduction section.
Comment 2
page 2 line 64 to 68. I'm not sure how these results can help a physician decide which modality to use for a particular patient? Your conclusion states that this still needs to be resolved in further studies.
Response>
Thank you for the kind comments. We definitely agree that it is still hard to decide which device should be used for a specific population. As we described already in Discussion section, this issue should be resolved in future longitudinal studies with solid end-points such as mortality.
Reviewer 2 Report
This article tried to find the inconsistency of skeletal muscle mass determined by CT and BIA, and wanted to find the predictors of discordance between these two methods. It attracted readers who are interested in body composition determination, and has novelty in this field.
My major concerns:
Sarcopenia is now a diagnosis in ICD-10. The diagnostic criteria are composed of low muscle mass, and either low grip strength or low gait speed. However, this manuscript is focused on low muscle mass, rather not sarcopenia. I strongly suggested that the authors should revise the term "sarcopenia" to "low muscle mass" in most situation. Please cite the following article for a better explanation. Association between Loss of Skeletal Muscle Mass, and Mortality and Tumor Recurrence in Hepatocellular Carcinoma: A Systematic Review and Meta-analysis. Liver Cancer 2017. DOI:10.1159/000484950
I am not sure Table 2 is correct? As authors stated, Table 2 is the comparison between low muscle mass and normal muscle mass subjects assessed by CT. The sex-specific cut-off value of LSMI were 46.12 in men and 34.18 in women (in Line 193). However, when I read the row of lumbar skeletal muscle index (LSMI) in Table 2, the range of nonsarcopenic is 23.2-85.6. A subject with LSMI 23.2 cannot be nonsarcopenic. In fact, the range should not be smaller than 34.18. Besides, the range of sarcopenic is 20.0-75.3. Again, the upper limit should not be larger than 46.12. Finally, since the cut-off value is determined by 2 SD below average, I guess the proportion of sarcopenia should be around 5%, not 15.1%. The authors should revise Table 2 and re-write the Result and the Discussion.
Minor issues:
Line110: 10000 x lumbar skeletal muscle area/height, please add "cm2" as the unit of height.
Line 137: Well written. I suggest cite "Skeletal muscle mass adjusted by height correlated better with muscular functions than that adjusted by body weight in defining sarcopenia. Sci Rep 2016; 6, 19457; doi: 10.1038/srep19457" for clearer explanation.
Line 166: 46.9kg/m2, should be 46.9cm2/m2.
Line 208: revision as "When BIA-define cutoff values were used, ..."
Line 218: ranged from -> was
Line 297: the prevalence of HTN and DM in the general Korean population were "lower" than this study... Not "higher"!
Author Response
My major concerns:
Comment 1
Sarcopenia is now a diagnosis in ICD-10. The diagnostic criteria are composed of low muscle mass, and either low grip strength or low gait speed. However, this manuscript is focused on low muscle mass, rather not sarcopenia. I strongly suggested that the authors should revise the term "sarcopenia" to "low muscle mass" in most situation. Please cite the following article for a better explanation. Association between Loss of Skeletal Muscle Mass, and Mortality and Tumor Recurrence in Hepatocellular Carcinoma: A Systematic Review and Meta-analysis. Liver Cancer 2017. DOI:10.1159/000484950
Response>
Thank you for the keen comments. We agree to the reviewer’s comment. Accordingly, we replace it with “low skeletal muscle mass” in most sentences.
Comment 2
I am not sure Table 2 is correct? As authors stated, Table 2 is the comparison between low muscle mass and normal muscle mass subjects assessed by CT. The sex-specific cut-off value of LSMI were 46.12 in men and 34.18 in women (in Line 193). However, when I read the row of lumbar skeletal muscle index (LSMI) in Table 2, the range of nonsarcopenic is 23.2-85.6. A subject with LSMI 23.2 cannot be nonsarcopenic. In fact, the range should not be smaller than 34.18. Besides, the range of sarcopenic is 20.0-75.3. Again, the upper limit should not be larger than 46.12. Finally, since the cut-off value is determined by 2 SD below average, I guess the proportion of sarcopenia should be around 5%, not 15.1%. The authors should revise Table 2 and re-write the Result and the Discussion.
Response>
Thank you for your careful comments. There were some mistakes. When cutoffs for below 2SD was used, the prevalence sarcopenia was only around 1%. Thus, we used 1SD cutoff for appropriate statistical analysis. Accordingly, we revised our manuscript. We are sorry again for the confusion.
Minor issues:
Comment 1
Line110: 10000 x lumbar skeletal muscle area/height, please add "cm2" as the unit of height.
Response>
Thank you for the comments. We revised.
Comment 2
Line 137: Well written. I suggest cite "Skeletal muscle mass adjusted by height correlated better with muscular functions than that adjusted by body weight in defining sarcopenia. Sci Rep 2016; 6, 19457; doi: 10.1038/srep19457" for clearer explanation.
Response>
Thank you very much for your kind advice. We added the reference.
Comment 3
Line 166: 46.9kg/m2, should be 1172.5px2/m2.
Response>
Thank you for the comments. We revised.
Comment 4
Line 208: revision as "When BIA-define cutoff values were used, ..."
Response>
Thank you for the comments. We modified the sentence.
Comment 5
Line 218: ranged from -> was
Response>
Thank you for the comments. We revised.
Comment 6
Line 297: the prevalence of HTN and DM in the general Korean population were "lower" than this study... Not "higher"!
Response>
Thank you for the comments. We revised.
Reviewer 3 Report
Thank you very much for the opportunity to review this manuscript. This study investigated the predictors for discordance between CT and BIA for muscle mass assessments. This paper is well written and constructed.
I have minor comments about this article.
Line 43, you mentioned that sarcopenia is officially recognized as a muscular disorder. However I checked the reference article 8, the authors didn’t state sarcopenia is a muscular disorder. They only mentioned it is a major cause of frailty and disability for elderly people. Please clarify.
Line 63, you mentioned that there is no gold standard for measuring skeletal muscle mass. In reference 2, the authors state that “CT and MRI are gold standards for estimating muscle mass in research.” It is different to your statement.
Line 95 you should cite reference about the measurement method.
The result and discussion part is well written.
In summary, this study provided information about the predictors of discordance of muscle mass evaluation by BIA and CT.
Author Response
Comment 1
Line 43, you mentioned that sarcopenia is officially recognized as a muscular disorder. However I checked the reference article 8, the authors didn’t state sarcopenia is a muscular disorder. They only mentioned it is a major cause of frailty and disability for elderly people. Please clarify.
Response>
Thank you for your comment. Because the term "Muscular disorder" might be confusing, we modified.
Comment 2
Line 63, you mentioned that there is no gold standard for measuring skeletal muscle mass. In reference 2, the authors state that “CT and MRI are gold standards for estimating muscle mass in research.” It is different to your statement.
Response>
Thank you for the keen comments. We revised the sentences to keep consistency in our statements.
Comment 3
Line 95 you should cite reference about the measurement method.
Response>
We have already cited reference #26 to show measurement methods.
Comment 4
The result and discussion part is well written.
Response>
Thank you very much for your kind reply.
This manuscript is a resubmission of an earlier submission. The following is a list of the peer review reports and author responses from that submission.
Round 1
Reviewer 1 Report
The authors evaluate and compare estimates of sarcopenia derived from measurements of muscle mass obtained from lumbar CT and whole-body BIA in apparently healthy adults. They report markedly greater prevalence of sarcopenia with CT than BIA. The authors should consider the following comments.
The topic of assessment of muscle mass and diagnosis of sarcopenia is timely. Importantly, ethnicity is a moderating factor that the authors also need to discuss. Specifically, two recent position papers should be considered: Dent E et al. J Nutr Health Aging 2018;22(10):1148-1161 and Cruz-Jentoft AJ et al. Age Ageing 2018 Oct 12. doi: 10.1093/ageing/afy169.
The description of methods to assess muscle mass requires additional discussion. The authors report that DXA is prone to overestimation of muscle mass in the presence of over-hydration or excess fluid accumulation. Similarly, BIA is prone to the similar problem. Thus, both methods suffer from the limitations of the two-component model that assumes constant level of hydration of the fat-free body to estimate body composition. How do the authors justify the use of BIA without assessment of hydration status?
A number of CT instruments are used to obtain the L3 scans. It is critical to describe the means to establish accuracy and precision of the measurements based on the diverse CT devices. Was a standardization trial performed to establish no difference among the L3 data from all of the CT instruments?
The authors should include additional information to establish the validity of the InBody BIA to estimate muscle mass; include reference(s). What is the accuracy and reproducibility of this device to estimate muscle mass? Clearly describe the regression equation(s) to predict appendicular and central muscle mass including the sources of error (e.g., SEE) and limits of agreement with the reference method. What are the principal independent predictors of muscle mass in this equation? If gender, age and body weight are the significant predictors, then the contribution of impedance variables, compared to these demographic variables, must be justified.
This discussion is important because age, gender and BMI are described as moderators in the Discussion. What percent of variance is attributable to these basic demographic characteristics and what is the specificity and sensitivity of each factor?
Both the European and International Sarcopenia Advisory Committees recommend the use of the validated BIA prediction equation of Janssen et (J Appl Physiol 2000; 89:465-471) for estimation of whole-body muscle mass for diagnosis of sarcopenia. Discuss the comparability of the InBody equation with the Janssen prediction model.
What is the previously established comparability of the CT determinations at L3 and the BIA estimates of muscle mass in diagnosis of sarcopenia? Does it differ among different ethnic groups, specifically European Caucasians and Asians?
Additional discussion of data distribution in panels of Figure 1 is needed. You report simple correlation coefficients but an analysis of the slope and intercept for comparability to line of identity is required. Are these relationships similar to similar analyses in the literature for different (non-Asian) ethnic groups?
Author Response
Point 1: The authors evaluate and compare estimates of sarcopenia derived from measurements of muscle mass obtained from lumbar CT and whole-body BIA in apparently healthy adults. They report markedly greater prevalence of sarcopenia with CT than BIA. The authors should consider the following comments.
The topic of assessment of muscle mass and diagnosis of sarcopenia is timely. Importantly, ethnicity is a moderating factor that the authors also need to discuss. Specifically, two recent position papers should be considered: Dent E et al. J Nutr Health Aging 2018;22(10):1148-1161 and Cruz-Jentoft AJ et al. Age Ageing 2018 Oct 12. doi: 10.1093/ageing/afy169.
Response 1: We reviewed two papers. According to 2018 European Working Group on Sarcopenia in Older People(EWGSOP), the BIA conversion equation is race-specific and it is based on DXA. Race is an important factor in the calculation of muscle mass through BIA. When using the BIA muscle mass prediction model according to the Sergi equation, it is difficult to apply to the Asian race because it is based on the european population. In addition, the prediction equation according to Asian race has not yet been fully defined yet. We refer to the above two papers and add them to the discussion.
In our study, the muscle mass was estimated by direct segmental multi-frequency bioimpedance analysis and we attach this information. Our research has been conducted on Asian races, so there may be restrictions on applying to all races.
Point 2: The description of methods to assess muscle mass requires additional discussion. The authors report that DXA is prone to overestimation of muscle mass in the presence of over-hydration or excess fluid accumulation. Similarly, BIA is prone to the similar problem. Thus, both methods suffer from the limitations of the two-component model that assumes constant level of hydration of the fat-free body to estimate body composition. How do the authors justify the use of BIA without assessment of hydration status?
Response 2: The hydration status is difficult to assess clearly, but the participants has been fasted for endoscopy and other tests. IThey are most likely the same hydration status with laxative taking for colonoscopy. We have defined the hydration status through these assumptions.
Point 3: The probability A number of CT instruments are used to obtain the L3 scans. It is critical to describe the means to establish accuracy and precision of the measurements based on the diverse CT devices. Was a standardization trial performed to establish no difference among the L3 data from all of the CT instruments?
Response 3: CT equipment is being standardized to minimize the possible differences for each individual device. We consulted with the Department of Radiology at our hospital. The muscle mass measurement is to measure the area in the CT image. It is not a concept to quantify the attenuation like the bone marrow density, so standardization does not go in. However, the quality of the image is constantly maintained by the equipment.
Point 4: The authors should include additional information to establish the validity of the InBody BIA to estimate muscle mass; include reference(s). What is the accuracy and reproducibility of this device to estimate muscle mass? Clearly describe the regression equation(s) to predict appendicular and central muscle mass including the sources of error (e.g., SEE) and limits of agreement with the reference method. What are the principal independent predictors of muscle mass in this equation? If gender, age and body weight are the significant predictors, then the contribution of impedance variables, compared to these demographic variables, must be justified.
Response 4: We have included additional information to demonstrate the validity of the InBody BIA. The accuracy and reproducibility of this device for measuring muscle masses is known to be high. In this study, we used the sum of direct muscle masses through direct segmental multi-frequency bioimpedance analysis rather than the muscle mass measurements described in the traditional regression equation. The amount of muscle mass was measured with reference to the following article. According to the Ann L Gibson et al. Am J Clin Nutr 2008;87:332–8 and Carolina H.Y. Ling et al. Clinical Nutrition 30 (2011) 610-615 papers, direct segmental multi-frequency bioimpedance analysis was statistically not significantly differences, when compared to muscle mass measured by DXA. According to a recently published study in Korean population, the muscle mass measured by direct segmental multi-frequency bioimpedance analysis was not significantly different from that measured with DXA. Seo Young Lee et al. Nutrients 2018, 10, 738; doi:10.3390/nu10060738.
In this study, we aimed to compare the muscle mass with the CT using a direct segmental multi-frequency bioimpedance analysis. The contents of direct segmental multi-frequency bioimpedance analysis are added to methods and introduction.
Point 5: This discussion is important because age, gender and BMI are described as moderators in the Discussion. What percent of variance is attributable to these basic demographic characteristics and what is the specificity and sensitivity of each factor?
Response 5: We will compare the age, sex, and body mass index of the subjects compared to the Korean population. The study population is composed of 79.3% of people over 50 years old and 41% of people over 60 years of age.
Point 6: Both the European and International Sarcopenia Advisory Committees recommend the use of the validated BIA prediction equation of Janssen et (J Appl Physiol 2000; 89:465-471) for estimation of whole-body muscle mass for diagnosis of sarcopenia. Discuss the comparability of the InBody equation with the Janssen prediction model.
Response 6: This study directly measured muscle mass through direct segmental multi-frequency bioimpedance analysis. We believe that it is important to verify the accuracy of direct segmental multi-frequency bioimpedance analysis by comparing the muscle mass measured with CT.
Point 7: What is the previously established comparability of the CT determinations at L3 and the BIA estimates of muscle mass in diagnosis of sarcopenia? Does it differ among different ethnic groups, specifically European Caucasians and Asians?
Response 7: The diagnostic criteria for sarcopenia differ between Europeans and Asians. This study was conducted with reference to EWGSOP and AWGS guidelines.
Point 8: Additional discussion of data distribution in panels of Figure 1 is needed. You report simple correlation coefficients but an analysis of the slope and intercept for comparability to line of identity is required. Are these relationships similar to similar analyses in the literature for different (non-Asian) ethnic groups?
Response 8: We will modify Figure1 to include comparable line and slope and intercept. This study is a study on Asian races. Based on our knowledge, similar topics were not found in other papers analyzing CT and BIA.
Reviewer 2 Report
General:
This is an interesting study that provides results that add to the scientific literature in this area. I feel there needs to be some reworking to get a better focus in the study and reduce the amount of data presented to the reader. I would like to see the work focussed on a primary objective and secondary objectives, each with a very clear research question/hypothesis and a clear statistical approach to address it. The discussion/conclusion should then follow logically. If it is felt necessary to keep all these analyses in the paper, I would suggest using supplementary material for some of the results.
Introduction:
1) Please provide some context around where this research fits with current literature. Other studies are referenced in the discussion, but I feel the introduction should have a short statement or two to highlight what has been done before and how this study aims to build on the previous work.
2) Page 2, line 56. “the assessment of skeletal muscle mass may be inaccurate if the patients are dehydrated or obese (11).” I would suggest changing ‘are dehydrated’ to have altered fluid status. The inaccuracies apply to overhydration as much as dehydration.
Methodology:
1) I would think carefully about whether two cutoffs are needed for each diagnostic test. Perhaps the most appropriate of the two could be used for the main analysis with a short sensitivity analysis done that highlights the main differences/similarities if the cutoff is changed. This would make a dramatic reduction in the readability and the amount of material presented in figures tables. Data from the second cutoffs could be added as supplementary material, with the salient points raised in the main text.
2) Please detail where Severance hospital is in the methods section and where the population came from. Also, can you comment on whether the reported prevalence of hypertension (41%) and diabetes (26%) is typical of an “apparently healthy” population in Korea?
Results:
1) Table 1: indent the CT and BIA results appropriately so the fibrosis-4 index could not be thought to come from the BIA data
2) Section 3.2, first paragraph. If I understand it correctly, the correlation coefficient between crude muscle mass by CT and BIA was 0.78 for the whole population, but split by gender, the coefficient fell to 0.5 in men and 0.4 in women?
3) Section 3.2, second paragraph. I don’t think this paragraph is necessary, the range for each quartile is on the graph and by definition, we know each quartile will be the given fraction of the total population.
4) Section 3.3, first line. Please replace “CT-based sarcopenia” by “CT-defined sarcopenia”
5) Section 3.4. I’m not sure it is useful to repeat all the results from table 3 in this paragraph (and section 3.3). Maybe refer the reader to the table and use the text to highlight anything of particular interest. E.g. gender was significantly different in one cutoff but not the other.
6) Section 3.5, second paragraph starts “To identify the predictors of discordant results by CT and BIA, univariate and subsequent 200 multivariate analysis were performed” but the rest of the paragraph presents results related to the prediction of sarcopenia, not discordant results.
Discussion:
1) When you talk about “discordance” I cannot see any indication of direction? Maybe it is present but there is so much information in the manuscript it is really difficult to digest. It would be nice to know if discordance meant higher prevalence of sarcopenia measured by BIA or by CT.
2) “All of these factors suggest that CT may be required for a more accurate assessment of muscle mass 244 in subjects with advanced age, female gender, and low BMI.” I don’t think this is necessarily a logical interpretation of the results, unless you can reference data that shows that diagnosis of sarcopenia by CT measured LSMI as compared to BIA is associated with any reference method or patient outcomes. The two techniques measure different things, as you state in the manuscript. How can one be seen to be more “accurate” than the other in diagnosing sarcopenia?
Author Response
We have carefully considered the valuable comments and suggestions provided by references and the editor, and made great efforts to improve the manuscript accordingly. The followings are point-by-point answers to specific questions raised by the reviewer. We hope that the revised version of manuscript could meet the priority required for the publication.
General:
This is an interesting study that provides results that add to the scientific literature in this area. I feel there needs to be some reworking to get a better focus in the study and reduce the amount of data presented to the reader. I would like to see the work focussed on a primary objective and secondary objectives, each with a very clear research question/hypothesis and a clear statistical approach to address it. The discussion/conclusion should then follow logically. If it is felt necessary to keep all these analyses in the paper, I would suggest using supplementary material for some of the results.
Response: To reduce the amount of data, we focused on primary and secondary aims and reduced the additional data as much as possible. And we will show some of the results through supplementary data.
Introduction:
1) Please provide some context around where this research fits with current literature. Other studies are referenced in the discussion, but I feel the introduction should have a short statement or two to highlight what has been done before and how this study aims to build on the previous work.
Response: We will add a little more content on the previous research to the introduction, and we will insert a sentence to emphasize the content of the research goal.
2) Page 2, line 56. “the assessment of skeletal muscle mass may be inaccurate if the patients are dehydrated or obese (11).” I would suggest changing ‘are dehydrated’ to have altered fluid status. The inaccuracies apply to overhydration as much as dehydration.
Response: Thank you for the appropriate intellectuals. We will change the contents of overhydration as well as dehydration.
Methodology:
1) I would think carefully about whether two cutoffs are needed for each diagnostic test. Perhaps the most appropriate of the two could be used for the main analysis with a short sensitivity analysis done that highlights the main differences/similarities if the cutoff is changed. This would make a dramatic reduction in the readability and the amount of material presented in figures tables. Data from the second cutoffs could be added as supplementary material, with the salient points raised in the main text.
Response: There are no definite criteria for diagnosis of sarcopenia, we wanted to analyze using two cut-off values. Therefore, there was a question about reliability when compared with only one cut-off value.
However, when considering your advice and considering the amount of data, we determined that it would be better to add a second cut-off values as a supplement.
2) Please detail where Severance hospital is in the methods section and where the population came from. Also, can you comment on whether the reported prevalence of hypertension (41%) and diabetes (26%) is typical of an “apparently healthy” population in Korea?
Response: In the case of hypertension and diabetes, data based on the participant's screening table are used.
Depending on the questionnaire, participants are asked to write past medical history, which may be inaccurate. The diagnosis of hypertension and diabetes may be overestimated by the health insurance system in Korea. People who receive health checkups are usually older than young people. When we analyzed the age of participants in this study, 79.3% of the participants were over 50 years old, and 41% were over 60 years old. Participants apparently have no problems with their activities and no problems due to serious illnesses. However, the prevalence of hypertension and diabetes is relatively high. We think it is because the participants are the elderly people.
Results:
1) Table 1: indent the CT and BIA results appropriately so the fibrosis-4 index could not be thought to come from the BIA data
Response: We will review the data and consider removal for the FIB-4 index.
2) Section 3.2, first paragraph. If I understand it correctly, the correlation coefficient between crude muscle mass by CT and BIA was 0.78 for the whole population, but split by gender, the coefficient fell to 0.5 in men and 0.4 in women?
Response: We have confirmed the statistics once again. For men and women, the statistical significance was the same but the correlation coefficient decreased. We estimate that the difference in sample numbers did not make a difference in the correlation coefficient.
3) Section 3.2, second paragraph. I don’t think this paragraph is necessary, the range for each quartile is on the graph and by definition, we know each quartile will be the given fraction of the total population.
Response: The authors thought that these figures could increase the reliability of the data. If you think that you have a lot of data and you are not readable, I will consider the change by attaching it to supplementary materials. The authors thought that these figures could increase the reliability of the data. If you think that you have a lot of data, we will consider the change by attaching it to supplementary materials.
4) Section 3.3, first line. Please replace “CT-based sarcopenia” by “CT-defined sarcopenia”
Response: Thank you for your comment and we will correct it.
5) Section 3.4. I’m not sure it is useful to repeat all the results from table 3 in this paragraph (and section 3.3). Maybe refer the reader to the table and use the text to highlight anything of particular interest. E.g. gender was significantly different in one cutoff but not the other.
Response: In case of repeated content, we will have reduced it to the minimum.
6) Section 3.5, second paragraph starts “To identify the predictors of discordant results by CT and BIA, univariate and subsequent 200 multivariate analysis were performed” but the rest of the paragraph presents results related to the prediction of sarcopenia, not discordant results.
Response: Thank you for your comment and we will correct it.
Discussion:
1) When you talk about “discordance” I cannot see any indication of direction? Maybe it is present but there is so much information in the manuscript it is really difficult to digest. It would be nice to know if discordance meant higher prevalence of sarcopenia measured by BIA or by CT
Response: Discordance indicates that the diagnosis of sarcopenia was different in CT and BIA. This was analyzed according to each cut-off value. For example, CT cut-off value indicates a diagnosis of sarcopenia but BIA cut-off value does not indicate a diagnosis of sarcopenia. Considering that there may be some confusion in the interpretation, I will add an additional explanation for the discordance.
2) “All of these factors suggest that CT may be required for a more accurate assessment of muscle mass 244 in subjects with advanced age, female gender, and low BMI.” I don’t think this is necessarily a logical interpretation of the results, unless you can reference data that shows that diagnosis of sarcopenia by CT measured LSMI as compared to BIA is associated with any reference method or patient outcomes. The two techniques measure different things, as you state in the manuscript. How can one be seen to be more “accurate” than the other in diagnosing sarcopenia?
Response: Our research has assumed that the muscle mass measured by CT is the most accurate way to measure the actual size directly. Therefore, we first assessed the statistical significance of the muscle mass measured by CT and the muscle mass measured by BIA. As with the results of the study, the BIA measurements showed a very similar relationship to the muscle mass measured by CT. As with the results of the study, there was not statistically significant difference in muscle mass measured by CT with BIA. There are some studies that compared the muscle mass of CT and BIA. However, these papers targeted specific groups of patients, such as patients with fatal or liver-related hospital experiences. The application to the general population has difficulties. Kim et al./Clinical Nutrition, Itoh S et al. Hepatol Res. 2016 Apr;46(4):292-7. doi: 10.1111/hepr.12537. Epub 2015 Jul 5.
Our study can not directly compare the muscle mass measured by CT with the muscle mass measured by BIA. So we have made a comparison through the diagnostic criteria of sarcopenia. Because of the high correlation between CT and BIA, we wanted to make sure that sarcopenia can be diagnosed simply through BIA. This will be very helpful in screening sarcopenia.